# The Survival of Septic Patients with Compensated Liver Cirrhosis Is Not Inferior to That of Septic Patients without Liver Cirrhosis: A Propensity Score Matching Analysis

**DOI:** 10.3390/jcm11061629

**Published:** 2022-03-15

**Authors:** Ya-Chun Chang, Ying-Tang Fang, Hung-Cheng Chen, Chiung-Yu Lin, Yu-Ping Chang, Yi-Hsuan Tsai, Yu-Mu Chen, Kuo-Tung Huang, Huang-Chih Chang, Chin-Chou Wang, Meng-Chih Lin, Wen-Feng Fang

**Affiliations:** 1Division of Pulmonary and Critical Care Medicine, Department of Internal Medicine, Kaohsiung Chang Gung Memorial Hospital, Chang Gung University College of Medicine, Kaohsiung 833, Taiwan; yachun1026@gmail.com (Y.-C.C.); cantico@pchome.com.tw (Y.-T.F.); chc1106@cgmh.org.tw (H.-C.C.); chiungyu@cgmh.org.tw (C.-Y.L.); b9002087@cgmh.org.tw (Y.-P.C.); flyninesun@gmail.com (Y.-H.T.); blackie@cgmh.org.tw (Y.-M.C.); jelly@cgmh.org.tw (K.-T.H.); kuan2101@yahoo.com.tw (H.-C.C.); ccwang5202@yahoo.com.tw (C.-C.W.); mengchih@cgmh.org.tw (M.-C.L.); 2Department of Internal Medicine, Kaohsiung Municipal Min-Sheng Hospital, Kaohsiung 802, Taiwan; 3Department of Respiratory Therapy, Kaohsiung Chang Gung Memorial Hospital, Chang Gung University College of Medicine, Kaohsiung 833, Taiwan; 4Department of Respiratory Care, Chang Gung University of Science and Technology, Chiayi 613, Taiwan

**Keywords:** sepsis, liver cirrhosis, intensive care, propensity score

## Abstract

Background: We aimed to determine whether septic patients with liver cirrhosis (LC) had worse survival than septic patients without liver cirrhosis (WLC). We also investigated the survival of septic patients with compensated liver cirrhosis (CLC) and decompensated liver cirrhosis (DLC). Methods: This study enrolled 776 consecutive adult patients with sepsis admitted to the medical intensive care units of a tertiary referral hospital. Clinical factors and laboratory data were collected for analysis. Propensity scoring was also used for the control of selection bias. The variables included in the propensity model were age, sex, presence of diabetes mellitus, hypertension, cardiovascular accident, chronic kidney disease, malignancy, APCHE II (Acute Physiology and Chronic Health Evaluation) score, hemoglobin, and platelet data on the day when sepsis was confirmed. Seven-day, ICU, and hospital mortality were analyzed after correcting for these confounding factors. Results: Of the 776 septic patients, 64 (8.2%) septic patients presented with LC. Patients were divided into two groups—LC (*n* = 64) and WLC (*n* = 712)—which presented different rates of hospital mortality (LC: 62.5% vs. WLC: 41.0%, *p* = 0.001). We further separated septic patients with LC into two groups: patients with CLC (*n* = 24) and those with DLC (*n* = 40). After propensity score matching, the survival of septic patients with CLC (63.6%) was not inferior to patients WLC (54.5%) (*p* = 0.411). Patients with DLC had more hospital mortality, even after matching (*p* < 0.05). The Quick SOFA (qSOFA) score, SOFA score, and sub-SOFA score were also comparable between groups. SOFA scores were not significantly different between the CLC and WLC groups after matching. Poor SOFA scores were observed in the DLC group on days 3 and 7 after matching (*p* < 0.05). Conclusions: Septic patients with LC had higher mortality compared to patients WLC before matching. However, after propensity score matching, the survival of septic patients with CLC was non-inferior to patients WLC.

## 1. Introduction

Sepsis, a life-threatening organ dysfunction, is caused by a dysregulated host response to infection; it is among the most common causes for admission to medical intensive care units (ICUs) [1,2] and demonstrates significant morbidity and mortality [3]. Septic patients with specific co-morbidities and risk factors are prone to poor outcomes [4,5]. However, the interplay between multiple organs in sepsis is complex [6]. For example, the liver plays important roles in immune response, coagulation, and detoxification [7]. Acute liver injury is one component of the Sequential Organ Failure Assessment (SOFA) score and is associated with mortality [8]. However, the SOFA score only collects total bilirubin data and does not encompass all information relevant to assessing the liver. In addition, the SOFA score does not evaluate the severity of liver disease.

The clinical course of cirrhosis has two phases [9]: compensated cirrhosis and decompensated cirrhosis. Compensated cirrhosis is defined as the period between the onset of cirrhosis and the first major complication, which is approximately 10 to 15 years. During this period, patients present with no or minor symptoms. However, if the etiological factor of cirrhosis persists, the histological liver lesions progress. Decompensated cirrhosis is when patients start to present with ascites, variceal hemorrhage or hepatic encephalopathy, spontaneous bacterial peritonitis and hepatorenal syndrome [10,11]. This period lasts approximately 3 to 5 years and is associated with a short survival time. Malfunction of the immune system at different levels is a known factor in patients with liver cirrhosis, and both non-specific and antigen-specific functions may be compromised. The diminished clearance capacity of the reticuloendothelial system in liver cirrhosis patients leads to a higher rate of bacterial infections [12]. However, the mortality of liver cirrhosis (LC) is not considered in patients with severe acute infection, such as patients with cirrhosis in a septic condition who require admission to the intensive care unit.

Studies on the influence of compensated liver cirrhosis (CLC) or decompensated liver cirrhosis (DLC) in patients with sepsis in medical intensive care units are limited. We aim to determine whether the survival of septic patients with LC is worse than patients without liver cirrhosis (WLC). We also investigate the survival of septic patients with CLC and DLC. We hypothesize that the DLC group may have worse outcomes compared to the group WLC. The outcomes of CLC may be similar to those for patients WLC.

## 2. Materials and Methods

### 2.1. Study Design, Participants and Setting

The study was part of an integrated research program. It consisted of a prospective observation study, including an immune profile study and retrospective medical chart review [13,14,15,16,17]. It was conducted in three medical ICUs at Kaohsiung Chang Gung Memorial Hospital (34 beds), a 2700-bed tertiary teaching hospital in southern Taiwan. We surveyed consecutive adult patients (aged ≥ 18 years) presenting with sepsis on admission to the medical ICU from August 2013 to January 2017 [18,19]. We excluded 23 patients who were re-admitted to the ICU during the study period. Among 23 patients re-admitted to the ICU, 3 patients with liver cirrhosis were re-admitted to the ICU and they were excluded from the study, as shown in Figure 1. We collected clinical data; laboratory data on the day that sepsis was confirmed and on admission days 1, 3, and 7; and outcomes in septic patients admitted to the medical ICUs during this period. The enrolled patients were divided into two groups: patients with LC (LC group) and patients WLC (WLC group). Septic patients with LC were further divided into two groups: patients with CLC (CLC group) and patients with DLC (DLC group) (Figure 1).

This study was approved by the Institutional Review Board of Chang Gung Memorial Hospital, and the requirements for patient consent were waived (202001498B0).

### 2.2. Definitions

LC is defined as a condition resulting from long-term damage of the liver [20]. This damage is characterized by the replacement of normal liver tissue with scar tissue. In the early phase, there are often no symptoms because the disease develops slowly, over months or years; this stage of disease is defined as CLC. Compensated cirrhosis is initially defined as the period between the onset of cirrhosis and the first major complication [21]. The concept of compensated advanced chronic liver disease (cACLD) was proposed by the Baveno VI consensus [22]. It was proposed to illustrate the range of severe fibrosis and cirrhosis. These patients do not have varices or ascites but should be monitored closely. A clearer role for transient elastography (TE) has been introduced. TE allows for the early detection of patients with CLD who may develop portal hypertension. Liver stiffness measured by TE is sufficient to suspect cACLD. If TE values are >15 kPa, then this is indicative of cACLD. If needed, the diagnosis of cACLD can be confirmed with liver biopsy, hepatic venous pressure gradient (HVPG), or upper endoscopy. Decompensated cirrhosis is defined when patients start to present with complications, such as hepatic encephalopathy, ascites or variceal hemorrhage, spontaneous bacterial peritonitis and hepatorenal syndrome. LC was diagnosed by medical chart record and ultrasound in our study. The features of liver cirrhosis in ultrasound were irregular liver surface, hypertrophy of the left segments, ascites, and signs of portal hypertension [23,24]. If the diagnosis of LC is questionable, alternative methods for the diagnosis of LC should be used, such as non-invasive and/or even invasive assessment of liver fibrosis.

According to the Third International Consensus Definitions for sepsis and septic shock, sepsis is defined as a life-threatening organ dysfunction due to a disproportionate host response to infection [8,25,26]. Organ dysfunction can be represented by an increase in SOFA score of 2 points or more, which is associated with an in-hospital mortality greater than 10%. Septic shock makes up a subset of patients with sepsis. It is identified as sepsis with persistent hypotension requiring vasopressors to maintain a mean arterial pressure ≥ 65 mm Hg despite adequate fluid resuscitation [26]. The SOFA score has six components [27]: (1) the respiratory sub-score (PaO_2_/FiO_2_ ratio and respiratory support); (2) coagulation sub-score (platelet count); (3) cardiovascular sub-score (mean arterial pressure and vasopressor support); (4) central nervous system sub-score (Glasgow Coma Score); (5) liver sub-score (total bilirubin); and (6) renal sub-score (creatinine). Each are rated 0 to 4 and summed into a final score from 0 to 24 [28]. Adult patients with suspected infection can be rapidly identified by a new bedside clinical score termed the Quick SOFA (qSOFA) score: respiratory rate of 22/min or greater, altered mentation, or systolic blood pressure of 100 mm Hg or less [26]. We screened all ICU patients with a qSOFA score of 2 points or greater and then we checked their SOFA score. We defined patients as having sepsis if their SOFA score increased by 2 points or greater from baseline and with suspected infection [8].

### 2.3. Data Collection

The following clinical data were retrieved from medical records: age, gender, underlying comorbidities [29,30], Acute Physiological Assessment and Chronic Health Evaluation II (APACHE II) score [31], SOFA score, qSOFA score, and SOFA sub-scores; laboratory data on the day when sepsis was confirmed and on admission days 1, 3 and 7, including white blood cell, hemoglobin, platelet count, total bilirubin, creatinine, albumin, and prothrombin time; 7-day, ICU, and hospital mortality; and other clinical factors possibly related to LC and sepsis outcomes. One of our authors manually reviewed the medical record for the listed complications. All septic patients with liver cirrhosis received ultrasound examination. We found the description of ascites by ultrasound record in order to further evaluate septic patients with compensated liver cirrhosis or decompensated liver cirrhosis.

### 2.4. Statistical Calculations

Categorical variables were analyzed using the Chi-squared test, and continuous variables were compared using the Mann–Whitney U test. A two-tailed *p* value < 0.05 was considered significant. Propensity scoring matching (PSM) was used for the control of selection bias and was performed using binary logistic regression to generate a propensity score for each patient who had or did not have LC. Variables included in the propensity model were age, sex, presence of diabetes mellitus, hypertension, cerebrovascular accident, chronic kidney disease, malignancy, APACHE II score during this admission, hemoglobin, and platelet data on the day when sepsis was confirmed. We selected these variables because they resulted in significant differences prior to propensity score matching. There were no significant differences among other laboratory data before propensity score matching. NCSS 10 statistical software (LLC, Kaysville, UT, USA) was used with the greedy algorithm for matching between the study groups with a 0.2 caliper width. The standardized mean difference (SMD) was used for the evaluation of covariate balance after PSM. Imbalance was defined as a standardized difference of >10%. After correcting for these confounding factors, 7-day, ICU, and hospital mortality analyses were repeated. All statistical analyses were performed using the SPSS 22.0 software package (IBM Corp. Released 2013. IBM SPSS Statistics for Windows, Version 22.0. Armonk, NY, USA: IBM Corp.).

## 3. Results

### 3.1. Patients’ Characteristics

A total of 776 patients were included in the study. The average age of this cohort was 68.0 (58.0–78.0) years, and the cohort was predominately male (60.2%) (Table 1). The LC group had 64 patients (8.2%). Physiological and laboratory characteristics in the WLC group and the CLC group are shown in Table 2. Most of the patients with DLC had hepatitis C virus infection (Table 3). Some cirrhotic patients had extrahepatic malignancies such as cholangiocarcinoma, colon cancer, esophageal cancer, etc. (Table 4). Some of them had stage IV cancer. However, the difference in cancer stage did not influence 7-day mortality, ICU mortality, or hospital mortality (7-day mortality: *p* = 0.753, ICU mortality: *p* = 0.405, and hospital mortality: *p* = 0.591).

Some of the patients with DLC had multiple comorbidities. However, the difference in number of comorbidities was not significant for 7-day mortality, ICU mortality, or hospital mortality (7-day mortality: *p* = 0.825, ICU mortality: *p* = 0.950, and hospital mortality: *p* = 0.817). No patients underwent liver transplantation during the study period. The reasons for ICU admission, besides sepsis, in the DLC group included two patients with hepatic encephalopathy, two patients with liver failure with hepatic coma, and five patients with hematemesis or gastrointestinal (GI) bleeding (Table 3).

After propensity score matching (PSM) for 10 variables, we found 57 matched patients in the LC group, 22 matched patients with CLC, and 33 matched patients with DLC (Appendix A). Seven-day mortality, ICU mortality, and hospital mortality were analyzed (Table 5). qSOFA score, SOFA score, and SOFA sub-scores were further analyzed in the WLC, LC, CLC, and DLC groups after matching (Appendix A).

### 3.2. Liver Cirrhosis Group vs. without Liver Cirrhosis Group

Patients with LC had worse rates for 7-day, ICU, and hospital mortality compared to patients WLC before PSM. Hospital mortality was 62.5% in the LC group and 41% in the WLC group (LC: 62.5% vs. WLC: 41.0%, *p* = 0.001) (Table 5).

### 3.3. Compensated Liver Cirrhosis Group vs. without Liver Cirrhosis Group

Septic patients showed no significant differences in 7-day, ICU, and hospital mortality between the CLC group and the WLC group before and after PSM (Table 5). The Kaplan–Meier curve of number of ICU days and hospital days in septic patients with CLC and WLC before and after PSM also resulted in no significant difference (Figure 2 and Figure 3). Therefore, the survival in septic patients with CLC was non-inferior to septic patients WLC.

### 3.4. Decompensated Liver Cirrhosis Group vs. without Liver Cirrhosis Group

The DLC group had worse outcomes in 7-day, ICU, and hospital mortality before PSM (Table 5). Even after PSM, septic patients with DLC had worse outcomes regarding hospital mortality (DLC: 75.8% vs. WLC: 45.5%, *p* < 0.05).

### 3.5. SOFA Scores and Sepsis-Related Inflammatory Markers

We further analyzed the qSOFA score, SOFA score, and SOFA sub-scores between the CLC, DLC and WLC groups after matching. There were no significant differences between the CLC and WLC groups in the qSOFA score, SOFA score, and SOFA sub-scores (*p* > 0.05). Poor SOFA scores were observed in the DLC group on days 3 and 7 after matching (*p* < 0.05) (Appendix A).

## 4. Discussion

The patients included were admitted to our tertiary teaching hospital, which also has a liver transplant center. We consulted the liver transplantation experts for further evaluation of whether the patient was suitable to receive a liver transplant. According to our results, septic patients with LC had a poor prognosis for 7-day, ICU, and hospital mortality before matching. However, after PSM, these patients did not have significantly worse 7-day, ICU, and hospital mortality (*p* > 0.05). For septic patients with CLC, the rates for 7-day, ICU, and hospital survival were non-inferior to the WLC group (Appendix A). After PSM, hospital mortality was significantly higher for the DLC group compared with that for the WLC group (DLC: 75.8% vs. WLC: 45.5%, *p* < 0.05). The Kaplan–Meier curve of ICU days and hospital days for septic patients in the CLC and WLC groups before and after PSM still revealed no significant differences (Figure 2 and Figure 3). The survival of septic patients with CLC was, thus, non-inferior to the WLC group after PSM analysis.

Sepsis can be considered as a fight between pathogens and the host’s immune system. It involves life-threatening organ dysfunction as a result of a dysregulated host response to infection [2,32]. Due to the high rate of mortality associated with the progression of organ dysfunction [33], some patients with sepsis face detrimental outcomes. Patients who have LC with sepsis are much less likely to survive [34]. Previous studies have observed similar results [35]. However, the study by Sauneuf et al., demonstrated that survival rates have increased in cirrhotic patients with sepsis in recent years [36]. They found that underlying liver disease appeared to be an important prognostic factor. Therefore, we need to further distinguish CLC and DLC, as the outcomes of these two conditions may be different.

Sauneuf et al., also reported that the implementation of therapeutic advances in sepsis possibly accounted for the observed increase in survival [36]. Based on our results, the survival of septic patients with CLC was non-inferior to septic patients WLC. Septic patients with DLC had a poor prognosis in terms of hospital mortality, even after PSM. Therefore, we encourage ICU intensivists to comply with sepsis campaign guidelines [37] to improve patient outcomes if the septic patient presents with CLC. If the septic patient presents with DLC, we recommend informing the patient’s family of the poor condition of the patient and setting the goals of care. According to the sepsis campaign guidelines, goals of care need to incorporate treatment and end-of-life care planning [37]. In addition, goals of care need to be addressed as early as is feasible.

One study has already revealed that malfunction of the immune system at different levels is typical for patients with LC [12]. The most important problem was the diminished clearance capacity of the reticuloendothelial system in patients with LC. This resulted in a significantly higher rate of bacterial infections associated with a poorer prognosis in these patients. This may be the reason why the DLC group with sepsis had poorer outcomes when compared to patients WLC, even after PSM. However, malfunction of the immune system, its mechanism, and the different levels of this in these patients with LC and sepsis still require further investigation.

Many researchers have studied the use of SOFA scores to predict infection-related in-hospital mortality in ICU patients [38,39]. The data show that the SOFA criteria were more effective in predicting mortality than the SIRS criteria and the qSOFA score. In addition, we also found some study groups already use Sepsis-3 criteria and qSOFA in patients with cirrhosis and bacterial infections [40,41]. In our study, we found that the qSOFA score, SOFA score, and SOFA sub-scores demonstrated no significant differences in the mortality of septic patients in the CLC and WLC groups after PSM (*p* > 0.05) (Appendix A). The stage of LC was more important in septic patients with LC. The DLC group had poorer SOFA scores on days 3 and 7 post-admission when compared to the WLC group after PSM (*p* < 0.05) (Appendix A). A poor SOFA score may be associated with higher hospital mortality in the DLC group.

This study did present some limitations. First, the study was a retrospective study in one tertiary referral teaching hospital. The disease of septic patients was more diverse and unpredictable (Table 3). Although all data for sepsis were collected prospectively, and all consecutive septic patients were enrolled, its retrospective nature and all the biases were derived from the retrieval of information from databases not designed for the study. The etiology of liver cirrhosis in 50–60% of patients was not indicated because of missing data or patients suffering from liver cirrhosis due to more than one etiology. Therefore, it is difficult to further analyze whether the etiology of the cirrhosis influences the prognosis. We also did not investigate whether there were differences in the distribution of etiologies of cirrhosis before and after propensity score matching. Second, the number of cases was relatively small, meaning selection bias was possible. In addition, the sample size was too small to detect significance between groups. This could be the reason for the insignificant results. However, all data for sepsis were collected prospectively, and all consecutive sepsis patients were enrolled; these features may facilitate a more complete analysis of the septic patient with LC. Third, not all of the WLC group received a liver ultrasound during hospitalization. Therefore, it is possible that some patients had undiagnosed cirrhosis. Fourth, our hospital has a liver transplant center. We were able to consult the liver transplantation team for further evaluation of whether the patient was suitable for receiving liver transplant. If the patient was suitable for receiving liver transplantation, they would be admitted or transferred to the surgical ward/surgical ICU for further evaluation. If the patient was not suitable for receiving liver transplantation, they were admitted to the medical ward/medical ICU for further management. Our study was conducted in medical ICUs in a renowned liver transplant center. We should be aware that it is possible that liver patients who present to non-liver transplant hospitals may not be as ill as those presenting to a transplant center.

## 5. Conclusions

Septic patients with LC had higher mortality compared to patients WLC before PSM. However, after PSM, we found that the survival of septic patients with CLC was non-inferior to septic patients WLC. Patients with DLC had significantly worse outcomes in terms of hospital mortality, even after PSM.

## Figures and Tables

**Figure 1 jcm-11-01629-f001:**
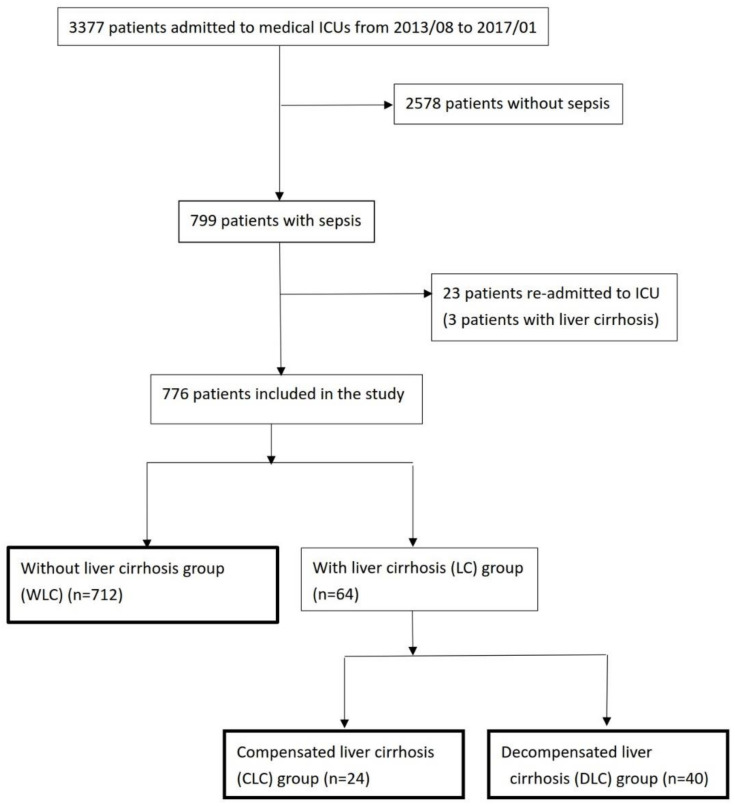
Among 799 patients who were diagnosed with sepsis from August 2013 to January 2017, 776 patients were included in the study. Abbreviations: WLC, without liver cirrhosis; LC, liver cirrhosis; CLC, compensated liver cirrhosis; DLC, decompensated liver cirrhosis.

**Figure 2 jcm-11-01629-f002:**
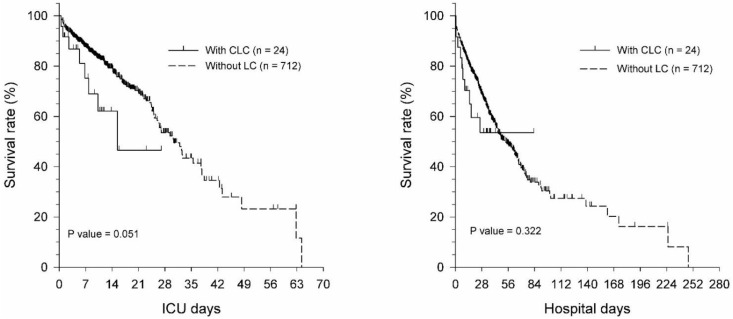
Kaplan–Meier curve of ICU days and hospital days for septic patients with CLC and WLC before propensity score matching. Abbreviations: LC, liver cirrhosis; CLC, compensated liver cirrhosis; WLC, without liver cirrhosis; ICU, intensive care unit.

**Figure 3 jcm-11-01629-f003:**
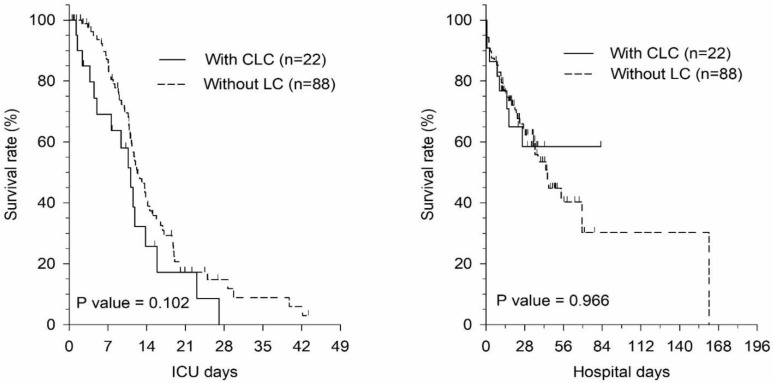
Kaplan–Meier curve of ICU days and hospital days for septic patients with CLC and WLC after propensity score matching. Abbreviations: LC, Liver cirrhosis; CLC, compensated liver cirrhosis; WLC, without liver cirrhosis; ICU, intensive care unit.

**Table 1 jcm-11-01629-t001:** Patient characteristics (*n* = 776 patients).

Factors	Overall *n* = 776	WLC *n* = 712	LC *n* = 64	CLC *n* = 24	DLC *n* = 40
Age, years, median (interquartile range)	68.0 (58.0–78.0)	69.0 (58.0–79.0)	62.0 (55.3–72.0)	61.0 (58.0–72.5)	62.0 (53.0–72.0)
Male sex, no. (%)	467 (60.2)	419 (58.8)	48 (75.0)	18 (75.0)	30 (75.0)
APACHE II, median (interquartile range)	24.5 (19.0–30.0)	24.0 (19.0–30.0)	25.5 (20.0–32.0)	26.5 (20.0–32.3)	25.0 (20.0–32.0)
7-day mortality, no. (%)	97 (12.5)	83 (11.7)	14 (21.9)	5 (20.8)	9 (22.5)
ICU mortality, no. (%)	205 (26.4)	178 (25.0)	27 (42.2)	8 (33.3)	19 (47.5)
Hospital mortality, no. (%)	332 (42.8)	292 (41.0)	(62.5)	10 (41.7)	30 (75.0)

Abbreviations: WLC, without liver cirrhosis; LC, liver cirrhosis; CLC, compensated liver cirrhosis; DLC, decompensated liver cirrhosis; ICU, intensive care unit; APACHE, Acute Physiology and Chronic Health Evaluation.

**Table 2 jcm-11-01629-t002:** Physiological and laboratory characteristics between the without liver cirrhosis group and the compensated liver cirrhosis group.

	WLC *n* = 712	CLC *n* = 24	*p* Value
**Vital signs, median (interquartile range)**			
Body temperature, degrees	36.9 (36.2–38.0)	36.5 (36.2–37.4)	0.435
Systolic pressure, mm Hg	129.0 (104.0–157.0)	124.0 (84.0–158.0)	0.498
Diastolic pressure, mm Hg	75.0 (61.0–89.0)	69.0 (57.8–89.8)	0.425
Mean artery pressure, mm Hg	94.8 (75.9–111.3)	88.5 (68.5–114.9)	0.416
**Laboratory data on the day sepsis was confirmed, median (interquartile range)**			
White blood cells, 10^6^/μL	12.6 (8.1–17.8)	11.0 (7.2–18.5)	0.713
Hemoglobin, g/dL	10.8 (9.1–12.5)	10.4 (9.3–12.7)	0.850
Platelet, 1000/μL	195.0 (114.7–271.0)	117.0 (45.5–196.0)	0.004
Creatinine, mg/dL	1.6 (1.0–3.0)	1.5 (1.0–3.5)	0.699
C-reactive protein, mg/L	114.2 (35.7–222.3)	82.5 (45.2–193.1)	0.776
Oxygenation index	6.7 (3.8–12.9)	9.1 (4.3–15.5)	0.156
**Clinical condition, no. (%)**			
Use of mechanical ventilation	662 (93.0)	24 (100.0)	0.179
Need for renal replacement therapy	130 (18.3)	4 (16.7)	0.842
**Site of lnfection *, no. (%)**			
Lung	468 (65.7)	13 (54.2)	0.242
Intra-abdomen	44 (6.2)	2 (8.3)	-
UTI	154 (21.6)	4 (16.7)	-
Bacteremia	53 (7.4)	2 (8.3)	-
Soft tissue or skin	34 (4.8)	2 (8.3)	-
Meningitis	3 (0.4)	1 (4.2)	-
Dengue fever	14 (2.0)	0 (0.0)	-
Influenza	5 (0.7)	0 (0.0)	-
Infective endocarditis	1 (0.1)	1 (4.2)	-
Unidentified	65 (9.1)	1 (4.2)	-

Abbreviations: WLC, without liver cirrhosis; CLC, compensated liver cirrhosis; UTI, urinary tract infection. * Some patients had more than one site of infection.

**Table 3 jcm-11-01629-t003:** Patient characteristics between the compensated liver cirrhosis group and the decompensated liver cirrhosis group.

	CLC *n* = 24	DLC *n* = 40
HBV, no. (%)	3 (12.5)	9 (22.5)
HCV, no. (%)	6 (25.0)	14 (35.0)
Alcoholic, no. (%)	2 (8.3)	4 (10.0)
Sum of above etiology, no. (%)	9 (47.5)	27 (67.5)
**Reasons for ICU admission, besides sepsis, no.**		
Acute respiratory failure	12	22
Out-of-hospital cardiac arrest	2	0
Asystole post-CPR	1	0
Massive hemoptysis	1	0
Pneumonia	3	1
Pulmonary embolism	0	1
Shock, except septic shock	0	1
Hepatic encephalopathy	0	2
Hepatic coma	0	2
Acute pancreatitis	1	1
Hematemesis or GI bleeding	0	5
Urinary tract infection	1	1
Dengue fever	0	1
Hemophagocytic syndrome	1	0
Influenza type A	0	1
Others	2	2

Abbreviations: HBV, Hepatitis B virus; HCV, Hepatitis C virus; CLC, compensated liver cirrhosis; DLC, decompensated liver cirrhosis; ICU, intensive care unit; CRP, C-reactive protein; GI, gastrointestinal.

**Table 4 jcm-11-01629-t004:** Malignancy types between the compensated liver cirrhosis group and the decompensated liver cirrhosis group.

Cancer Types	CLC *n* = 8	DLC *n* = 13
Cholangiocarcinoma	0	1
Colon cancer	1	2
Esophageal cancer	1	1
Hepatocellular carcinoma	3	9 ^a,b,c^
Breast cancer	0	1 ^a^
Nasopharyngeal cancer	1	0
Hypopharyngeal cancer	1	0
Lung cancer	1	0
Tongue cancer	0	1 ^b^
Non-Hodgkin’s lymphoma	0	1 ^c^
Bladder cancer	0	1 ^c^

Abbreviations: CLC, compensated liver cirrhosis; DLC, decompensated liver cirrhosis. ^a^ One patient had breast cancer and hepatocellular carcinoma. ^b^ One patient had tongue cancer and hepatocellular carcinoma. ^c^ One patient had non-Hodgkin’s lymphoma, hepatocellular carcinoma, and bladder cancer.

**Table 5 jcm-11-01629-t005:** Mortalities were analyzed in the without liver cirrhosis group vs. the liver cirrhosis and without liver cirrhosis groups vs. the decompensated liver cirrhosis group before and after matching.

	Before Matching			After Matching			
WLC vs. LC, no. (%)	**WLC** ***n* = 712**	**LC** ***n* = 64**	** *p* ** **Value ^a^**	**WLC** ***n* = 114**	**LC** ***n* = 57**	**Adjusted OR**	** *p* ** **Value ^b^**
7-day mortality	83 (11.7)	14 (21.9)	0.018	20 (17.5)	13 (22.8)	1.42 (0.63–3.20)	0.398
ICU mortality	178 (25.0)	27 (42.2)	0.003	41 (36.0)	24 (42.1)	1.30 (0.67–2.50)	0.435
Hospital mortality	292 (41.0)	40 (62.5)	0.001	56 (49.1)	35 (61.4)	1.71 (0.87–3.36)	0.121
WLC vs. CLC, no. (%)	**WLC** ***n* = 712**	**CLC** ***n* = 24**	** *p* ** **value ^a^**	**WLC** ***n* = 88**	**CLC** ***n* = 22**	**Adjusted OR**	** *p* ** **value ^b^**
7-day mortality	83 (11.7)	5 (20.8)	0.173	12 (13.6)	3 (13.6)	1.000 (0.24–4.11)	1.000
ICU mortality	178 (25.0)	8 (33.3)	0.355	26 (29.5)	6 (27.3)	0.895 (0.32–2.53)	0.835
Hospital mortality	292 (41.0)	10 (41.7)	0.949	40 (45.5)	8 (36.4)	0.646 (0.23–1.83)	0.411
WLC vs. DLC, no. (%)	**WLC** ***n* = 712**	**DLC** ***n* = 40**	** *p* ** **value ^a^**	**WLC** ***n* = 99**	**DLC** ***n* = 33**	**Adjusted OR**	** *p* ** **value ^b^**
7-day mortality	83 (11.7)	9 (22.5)	0.042	20 (20.2)	8 (24.2)	1.26 (0.50–3.16)	0.628
ICU mortality	178 (25.0)	19 (47.5)	0.002	33 (33.3)	16 (48.5)	1.81 (0.83–3.95)	0.136
Hospital mortality	292 (41.0)	30 (75.0)	<0.001	45 (45.5)	25 (75.8)	4.61 (1.66–12.82)	0.003

Abbreviations: WLC, without liver cirrhosis; LC, liver cirrhosis; CLC, compensated liver cirrhosis; DLC, decompensated liver cirrhosis; OR, odds ratio; ICU, intensive care unit. ^a^ Binary logistic regression. ^b^.Conditional logistic regression.

## Data Availability

The data presented in this study are available upon request from the corresponding author.

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
