# Peer review of "The Survival of Septic Patients with Compensated Liver Cirrhosis Is Not Inferior to That of Septic Patients without Liver Cirrhosis: A Propensity Score Matching Analysis"

_jcm, 2022, doi:10.3390/jcm11061629_

Round 1
Reviewer 1 Report
This is an interesting manuscript about a problem of clinical relevance, sepsis in patients with cirrhosis. The manuscript is well written and interesting results are shown. However, several items need to be clarified:
- How many patients re-admitted to ICU and excluded of study had liver cirrhosis?
- In table 3 is not indicated the etiology of 50-60% of patients with cirrhosis. Why? Could the etiology of the cirrhosis have influenced the prognosis? Were there differences in the distribution of etiologies of cirrhosis before and after propensity score matching?
- The main limitation of the study is its retrospective nature and all the biases derived from the retrieval of information from databases not designed for the study. Have the authors evaluated the quality of the records in terms of information collected/loss of information and veracity?
Reviewer 2 Report
Chang et al tackle an important topic, the survivability of sepsis by cirrhotic patients. Their article is overall a very useful addition to the literature. However, I worry that they have not clearly explained how patients were categorized in the WLC, CLC, and DLC categories and how patients were included in the study in the first place. In addition, I fear they have not controlled for the severity at presentation adequately.
Introduction: Well written and informative. Would add a short paragraph on the effect of cirrhosis on immune system function, since the article talks about survivability of sepsis.
Methods:
- Please discuss specifically how cirrhosis was identified radiographically (based on what US criteria)
- Please explain whether every patient with sepsis had an ultrasound of the liver to classify into LC or WLC groups or whether US was obtained based on clinical judgement. If there were patients in the study that did not receive US, there is serious potential for bias
- Please explain how it was determined whether the patient had DLC or CLC. Did one of the authors manually review the medical record for the listed complications? Was this done through an electronic “chart search” procedure? Through reviewing of specific notes?
- Please discuss whether SBP and hepatorenal syndrome should be included in the definition of DLC
- Please discuss how you decided which of the ICU patients met the criteria for sepsis. The authors describe SOFA and qSOFA but don’t discuss which exact criteria they used.
- Please discuss how you selected covariates for the propensity score matching model
Results
- I worry that the authors fail to control for crucial variables in their propensity score matching. I think it would be critical to control for type and severity of infection causing sepsis and the severity of the septic shock at presentation.
- Tables 5-7 can be delegated to the supplement
Discussion
- I think it is worth discussing whether the authors’ institution is a liver transplant center or not. If not, it is possible that the liver patients who present there are not as sick as those presenting to a transplant center. In general, it would be worth devoting some space in the discussion to how patients are triaged to this hospital/ICU and whether bias could enter in that decision.
Round 2
Reviewer 2 Report
The authors have addressed my concerns. I have one remaining concern: Did the WLC group have a liver US during this hospitalization? How did the authors know that those patients did not have cirrhosis? Did they base that on review of the medical record? If so, please state that and discuss it as a limitation of the study (it is possible that some patients had undiagnosed cirrhosis).
Additionally, significant English language editing is required.
